# House Flies (*Musca domestica*) Pose a Risk of Carriage and Transmission of Bacterial Pathogens Associated with Bovine Respiratory Disease (BRD)

**DOI:** 10.3390/insects10100358

**Published:** 2019-10-18

**Authors:** Saraswoti Neupane, Dana Nayduch, Ludek Zurek

**Affiliations:** 1Department of Entomology, Kansas State University, Manhattan, KS 66502, USA; 2Arthropod-borne Animal Diseases Research Unit, USDA-ARS, 1515 College Avenue, Manhattan, KS 66502, USA; 3Department of Pathology and Parasitology, Center for Zoonoses, University of Veterinary and Pharmaceutical Sciences, Central European Institute of Technology, 61242 Brno, Czech Republic; 4Department of Chemistry and Biochemistry, Mendel University, 61300 Brno, Czech Republic

**Keywords:** bovine respiratory disease, house fly, bacterial pathogen, cattle

## Abstract

House flies are important nuisance pests in a variety of confined livestock operations. More importantly, house flies are known mechanical vectors of numerous animal and human pathogens. Bovine respiratory disease (BRD) is an economically important, complex illness of cattle associated with several bacteria and viruses. The role of flies in the ecology and transmission of bacterial pathogens associated with BRD is not understood. Using culture-dependent and culture-independent methods, we examined the prevalence of the BRD bacterial complex *Mannheimia haemolytica, Pasteurella multocida* and *Histophilus somni* in house flies collected in a commercial feedlot from a pen with cattle exhibiting apparent BRD symptoms. Using both methods, *M. haemolytica* was detected in 11.7% of house flies, followed by *P. multocida* (5.0%) and *H. somni* (3.3%). The presence of BRD bacterial pathogens in house flies suggests that this insect plays a role in the ecology of BRD pathogens and could pose a risk as a potential reservoir and/or a vector of BRD pathogens among individual cattle and in their environment.

## 1. Introduction

Bovine respiratory disease (BRD) is a major problem in feedlot cattle production in the USA, and outbreaks can result in significant economic losses [1,2]. The onset of disease is primarily linked to environmental stressors such as shipping, crowding and changes in diet [3] that presumably cause immunosuppression and subsequent infections from various viral and bacterial pathogens [2,4]. The bacteria associated with BRD, *Mannheimia haemolytica, Pasteurella multocida, Histophilus somni*, and *Mycoplasma bovis*, are routinely isolated from cattle with BRD [5,6,7]. The prevalence of these bacteria in cattle production systems is highly variable [8]. Most studies have focused on management strategies to avoid disease and on detection, characterization, drug resistance and drug development for these pathogens, but there is little information about pathogen ecology and transmission dynamics.

House flies are among the most common insect pests in confined animal facilities, including cattle feedlots. Larvae develop in microbe-rich environments such as those contaminated with animal feces. Adult flies frequent these locations and readily feed upon septic substrates (e.g., manure), as well as animal wounds, secretions, food and water [9]. Therefore, adult flies serve as both reservoirs and vectors of microbes, including many bacterial pathogens [10]. House flies carry bacteria in their digestive tract and on the body surface and disseminate/release them by regurgitation from the foregut, by defecation, and by dislodging bacterial cells from the body surface, primarily from their tarsi [10]. While house flies have been implicated in the carriage and transmission of other pathogens of cattle, including some that are zoonotic [11,12,13,14], their association with BRD bacteria or their role in the transmission of BRD bacteria has not been examined to date.

In this study, we aimed to assess the prevalence of the BRD-associated bacterial pathogens *M. haemolytica, P. multocida* and *H. somni* in house flies from a commercial cattle feedlot with animals suffering from BRD symptoms using culture-dependent and culture-independent methods.

## 2. Materials and Methods

### 2.1. Bacterial Isolates and Culture Methods

Pure cultures of BRD-associated bacterial pathogens *M. haemolytica* DM1 and *P. multocida* DM1 were received from Dr. Derek Mosier, Kansas State University, while *H. somni* B55 was obtained from Dr. Brian Lubbers, Kansas State University. *M. haemolytica* and *P. multocida* were cultured on brain heart infusion agar (BHI; BD, Franklin Lakes, NJ, USA) supplemented with 5% sheep blood (Cedarlane^®^, Burlington, NC, USA) and antimicrobials (per ml: 5 µg neomycin, 5 µg vancomycin, 50 µg sodium azide and 100 µg cycloheximide) and used as a reference (positive control) for both culture-dependent and -independent methods. The bacteria were incubated at 37 °C for 24 h. *H. somni* was cultured on chocolate agar (CA) supplemented with 5% sheep blood and antimicrobials as above. Bacteria were incubated for 48 h at 37 °C in a BBL GasPak system (Becton, Dickinson and Company, Franklin Lakes, NJ, USA) containing a CampyPak Plus™ microaerophilic system generator (Becton, Dickinson and Company, Franklin Lakes, NJ, USA). These bacterial isolates were also used as positive controls.

### 2.2. Field Site, Sample Collection and Processing

A commercial cattle operation located in Reading, KS (38°37′31″ N, 95°58′23″ W), was chosen as the collection site on the basis of information obtained from the owner which indicated the presence of cattle with respiratory health issues. House flies were collected two times from an area outside an open-air fenced sick-cattle pen at the feedlot. On the first sampling date (13 June 2018), 30 house flies (12 males and 18 females) were captured by sweep, immediately transferred to a sterile Ziploc^®^ bag, kept cool on ice during transport to the laboratory, and processed within ~4 h. On the second sampling date (26 June 2018), 30 house flies (18 males and 12 females) were captured from the same area and processed as above.

In the laboratory, each fly was homogenized in 500 µL sterile phosphate-buffered saline (PBS) (pH 7.2; MP Biomedicals, LLC, Irvine, CA, USA). Two aliquots of fly homogenate were prepared for the subsequent detection of bacteria: a 200 µL aliquot was used for culture-dependent analysis (pathogen culture), and the remaining 300 µL was stored at −80 °C for detection of pathogens using a culture-independent method (DNA extraction, PCR and sequencing).

### 2.3. Culture-Dependent Method

A fly homogenate (100 µL) was plated on blood agar, and another aliquot (100 µL) on chocolate agar medium supplemented with antibiotics and cultured as described above. Due to constraints of microaerophilic conditions, we plated only 54 fly homogenates on chocolate agar for culturing of *H. somni*. Colonies with morphology similar to those of the reference isolates *M. haemolytica* and/or *P. multocida* and *H. somni* (Klima et al. [5]) were sub-cultured on blood or chocolate agar, respectively. Colonies exhibiting similar morphologies were enumerated to estimate their abundance (colony-forming units, CFU, per fly). Species identification was confirmed by PCR amplification and sequencing as described below.

### 2.4. DNA Preparation, PCR and Sequencing of Cultured Bacteria

A single colony (selected isolates and reference isolates) was picked from a pure culture and suspended in 100 µL of nuclease-free water, boiled for 15 min at 100 °C and centrifuged at 10,000× *g* for 3 min. The supernatant was used as a DNA template for PCR. Species-specific primers HS [5], Leukotoxin A [15] and KMT1_2 (hydrolase family) [16,17] were chosen for *H. somni*, *M. haemolytica* and *P. multocida* identification, respectively (Table 1). Further, a universal bacterial primer pair targeting the 16S rRNA gene (8F and 806R) [18,19] was used as a reaction control. PCR assays with single pairs of primer sets were carried out using DNA extracted from selected bacterial isolates; reference bacterial isolates were used as positive controls, and nuclease-free water was used as a negative reaction control.

In addition, the species-specific primer pairs were optimized for the detection of bacterial pathogens directly in the fly homogenates spiked with a reference bacterial culture (positive controls). The DNA was extracted and amplified using species-specific primer pairs for *H. somni*, *M. haemolytica* and *P. multocida*, as described below. The detection levels of the specific primers were 60 CFU/µL, 180 CFU/µL and 115 CFU/µL for *H. somni*, *M. haemolytica* and *P. multocida,* respectively.

Each PCR reaction (25 µL) contained a final concentration of 1× PCR Master Mix (Promega, Madison, WI, USA), 0.2 µM of each primer (Bio-Synthesis Inc, Lewisville, TX, USA) and 2.5 µL of DNA template. The PCR was performed using a DNA Engine^®^ Thermal Cycler (Bio-Rad, Hercules, CA, USA), with the following reaction conditions: 94 °C for 3 min, followed by 25 cycles at 94 °C for 30 s, 55 or 58 °C (Table 1) for 30 s and 72 °C for 30 s; then, 72 °C for 5 min. To confirm amplification, 5 µL of PCR product was visualized on 1% agarose gel via gel electrophoresis. Amplicons were purified using DNA Clean and Concentrator-5 kit (Zymo Research Corporation, Irvine, CA, USA) as described in the manufacturer’s instructions. Purified amplicons were sequenced using the Sanger sequencing method by Genewiz (https://www.genewiz.com/en, GENEWIZ, LLC, South Plainfield, NJ, USA). The sequences were compared with those in the National Center for Biotechnology Information non-redundant reference dataset using a basic local alignment search tool [20]. Sequence identity was confirmed when a sequence was ≥99% similar to a reference species.

### 2.5. Culture-Independent Method

Total genomic DNA was extracted from 300 µL of fly homogenate using the ZymoBIOMICS DNA kit (Zymo Research Corporation, Irvine, CA, USA) and amplified using pathogen-specific primer pairs for *H. somni* (HS), *M. haemolytica* (Leuk-A) and *P. multocida* (KMT1-2) (Table 1). The same PCR reaction conditions as those reported above were used, and amplicon purification, sequencing and analysis were performed as described above.

## 3. Results

### Prevalence of M. haemolytica, P. multocida and H. somni in House Flies from a BRD-Affected Commercial Feedlot

In total, 60 individual house flies were processed, and the homogenates were cultured to detect *M. haemolytica* and *P. multocida*, while 54 individuals were processed for culturing of *H. somni*. *M. haemolytica* was isolated from two female house flies, whereas *P. multocida* was isolated from three females. Similarly, *H. somni* was isolated from one female out of 54 house flies, and CFUs were too numerous to count (TNTC; Table 2). The number of CFUs of *M. haemolytica* ranged between 125 and 1000 CFUs/fly, while those of *P. multocida* ranged between 310 and 1000 CFUs/fly (Table 2). All culture-positive flies were females and were collected during the first sampling date.

The culture-independent direct PCR method revealed that out of the total 60 house flies, 4 flies from the first sampling date and two flies from the second sampling date were positive for *M. haemolytica* (Table 3), and this was confirmed by sequencing. However, no house flies were PCR-positive for *P. multocida*. Two out of 60 house flies were positive for *H. somni* by PCR and sequencing (Table 3), and both samples were collected on the first sampling date. Taken together, the culture-dependent and -independent methods detected at least one of the BRD pathogens in eight house flies, of which seven were females: all three pathogens from one female fly and two of three pathogens from two female flies. Using either method, the overall prevalence of *M. haemolytica*, *P. multocida* and *H. somni* in house flies collected from the commercial feedlot was 11.7%, 5.0% and 3.3%, respectively.

## 4. Discussion

The BRD pathogen complex includes *M. haemolytica*, *P. multocida, H. somni, Mycoplasma bovis*, bovine respiratory syncytial virus, and bovine herpesvirus-1 [2]. The bacterial BRD pathogens can be found in the respiratory tract of both sick and healthy cattle [21,22]. The disease onset is linked to various stressors such as weather conditions [23] and transportation [24]. Such stressors may cause immunosuppression in cattle that ultimately contributes to the respiratory tract microbiota becoming pathogenic [2,4]. Transmission of BRD pathogens among cattle is considered to occur via direct contact with infected animals [25] or by ingesting feed and water contaminated with nasal discharge containing these bacteria; however, transmission via insect vectors has not been explored previously. Our study focused on determining the prevalence of the three main bacterial BRD pathogens in house flies, one of the primary arthropod pests of confined livestock operations. Using both culture-based and PCR-based methods, we demonstrated that >3% of house flies from a BRD-affected commercial feedlot carried at least one of the BRD-associated bacterial pathogens (i.e., *M. haemolytica*, *P. multocida* or *H. somni*). This suggests that house flies could serve a role in BRD outbreaks by harboring and potentially transmitting BRD bacterial pathogens.

The culture-based method revealed that a high percentage of female house flies (16.7% of total females) carried at least one of the BRD bacterial pathogens. Thus, our preliminary findings suggest that females may pose a higher risk of dissemination of viable bacteria than male flies. Also, when the culture-based and direct PCR methods were combined, seven of eight (87.5%) house flies that possessed any one of these pathogens were females. Sex-specific behaviours of house flies may compel females to feed on more proteinaceous substances and to interact more frequently with microbe-rich substrates than males, due to their reproductive needs (i.e., anautogeny, oviposition sites) [26]. We, therefore, speculate that as female flies more frequently feed on microbe- and nutrient-rich substrates in a feedlot, such as manure, animal secretions, wounds, animal feed, etc., they become contaminated with pathogens shed from infected animals. Indeed, *M. haemolytica*, *P. multocida* and *H. somni* are routinely cultured from tonsillar tissues and lung tissues of infected (sick) cattle and can be particularly abundant in nasal secretions [5,7,15,27]. Consequently, house flies likely acquire BRD pathogens by feeding on either nasal secretions or contaminated feed and water. Once flies are contaminated with bacterial pathogens, either on their surface or by ingestion, they can disseminate these bacteria into a range of new environments, including locations with healthy cattle. However, since we homogenized the whole flies, we could not differentiate whether the BRD bacteria originated from the fly digestive tract, from the body surface or from both. More detailed studies focused on the vectorial capacity of house flies for BRD pathogens are warranted. 

We also found that the prevalence of these bacteria in this feedlot seemed to be associated with the presence of and duration of sick cattle in the sick pen. On the first sampling date, when all three bacterial pathogens were isolated, the pen contained sick calves that had been there for several days. In contrast, on the second sampling date, new sick calves had arrived on the day of collection. To fully understand the role of house flies in the ecology of these microbes, from acquisition to harbouring and to dissemination, it will be paramount in future studies to also measure the prevalence and persistence of BRD pathogens in relation to the infected animals (and their secretions) as well as their environment (feed, water, fomites).

Our study used both culture-dependent and -independent methods to successfully detect BRD bacterial pathogens in house flies. There are benefits and caveats concerning the use of either approach, as demonstrated previously by Gupta et al. [28] and in our study. Our results demonstrated limitations of each technique, since in some instances, bacteria were cultured but not detected by PCR (e.g., in three flies for *P. multocida* and one fly for *M. haemolytica*), and PCR detected bacterial DNA but the culturing was unsuccessful (e.g., in five flies for *M. haemolytica* and one fly for *H. somni*). These discrepancies could be due to the limitations of both methods to detect specific bacterial pathogens from the complex bacterial communities in the house fly homogenate. Culture-dependent methods target viable bacteria in the fly homogenate, which must out-compete other diverse microorganisms on culture plates in order to be successfully detected and enumerated. Thus, failure of bacteria detection by culture can be attributed to the fastidious nature of many of the target pathogens as well as to antagonistic and competitive actions of other bacteria in the sample. In addition, if the target pathogens are in relatively low abundance compared to other bacteria, then sub-sampling of the homogenate (i.e., plating a sample rather than the whole fly homogenate) can result in under-detection due to abundance below the limits of detection for the assay. Successful culture-independent methods similarly rely on the abundance of the target bacterial cells in the sample and successful DNA extraction followed by amplification of the targeted gene by PCR. While PCR is generally considered to be more sensitive than culture-dependent approaches [28], unsuccessful DNA isolation as well as the low copy number of the target genes can result in false-negative results. Additionally, PCR does not discriminate between viable and unviable bacteria, which is important to consider in terms of the potential spread of a disease.

## 5. Conclusions

In conclusion, we demonstrated that house flies collected in the close proximity of cattle with apparent symptoms of BRD harbored *M. haemolytica*, *P. multocida* and *H. somni*. The prevalence of these bacteria in house flies (>3%) indicates that flies serve as a potential reservoir and/or a vector for these pathogens within feedlots and surrounding areas. Further studies are required to fully understand the vector competence of house flies for BRD pathogens and the extent of risk that house flies, especially females, pose for harbouring and transmitting BRD in the field.

## Figures and Tables

**Table 1 insects-10-00358-t001:** PCR primers for bacterial species confirmation.

Bacterium	Primer Name (s)	Forward 5′–3′	Reverse 5′–3′	Annealing Temperature (°C)	Fragment Size	Reference
*Histophilus somni*	HS	GAAGGCGATTAGTTTAAGAG	TTCGGGCACCAAGTRTTCA	55	400	[5]
*Mannheimia haemolytica*	Leuk-A	CTTACATTTTAGCCCAACGTG	TAAATTCGCAAGATAACGGG	58	497	[15]
*Pasteurella multocida*	KMT1_2	GTGTGTTGAGCCAATCTGCT	GCTGTAAACGAACTCGCCAC	55	283	[16,17]
Bacteria	8F, 806R	AGAGTTTGATCCTGGCTCAG	GGACTACCAGGGTATCTAAT	52	799	[18,19]

**Table 2 insects-10-00358-t002:** Respiratory disease (BRD)-associated bacteria isolated from house flies from a BRD-affected feedlot.

FlyID	Fly Sex ^a^	Bacterium	CFU/fly *
**L1F3**	Female	*M. haemolytica*	~1000
**L1F13**	Female	*M. haemolytica*	125
**L2F2**	Female	*P. multocida*	1000
**L1F8**	Female	*P. multocida*	750
**L1F2**	Female	*P. multocida*	310
**L1F2**	Female	*H. somni*	TNTC *

* Colony-forming units (CFUs) were too numerous to count (TNTC) by the culture method used. ^a^ All BRD-associated bacteria were isolated from house flies collected on the first sampling date.

**Table 3 insects-10-00358-t003:** Prevalence (+) of *M. haemolytica, P. multocida* and *H. somni* in house flies from a bovine respiratory disease-affected feedlot.

FlyID	Fly Sex ^a^	Sampling Date ^b^	*M. haemolytica*		*P. multocida*		*H. somni*		Bacteria
			Culture	Direct PCR	Culture	Direct PCR	Culture	Direct PCR	PCR
L1F2	F	1		+	+		+	+	+
L1F3	F	1	+	+				+	+
L1F8	F	1		+	+				+
L1F13	F	1	+						+
L2F2	F	1			+				+
L2F4	F	1		+					+
L3F29	M	2		+					+
L3F30	F	2		+					+

^a^ Fly sex F = female, M = male. ^b^ Sampling Date: 1 = First sampling date, 2 = Second sampling date.

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
