# Peer review of "House Flies (Musca domestica) Pose a Risk of Carriage and Transmission of Bacterial Pathogens Associated with Bovine Respiratory Disease (BRD)"

_insects, 2019, doi:10.3390/insects10100358_

Round 1

Reviewer 1 Report

The study assessed the prevalence of BRD pathogens in house flies sampled from a sick-cow holding facility.  

Two approaches at bacterial identification were employed, and it was determined that a small percentage of sampled flies did contain at least one of the BRD-associated pathogens.  

While the prevalence of pathogens in sampled flies was low, the study represents a first step in establishing a role for houseflies in BRD ecology.  

As the samples were based on homogenates, the study does not indicate whether the pathogens were on the exterior of the flies or if they were located in the alimentary tract.  Establishing whether or not the pathogens could be deposited vial tarsal contact/regurgitation/defecation would greatly increase the likelihood of a role for M. domestica in BRD transmission.  Perhaps including a discussion of these limitations in would increase the value of this report to future researchers. 

Author Response

Dear reviewer,

thank you for your constructive comments. We have added a comment on the limitation in our study regarding the localization of the BRD pathogen in/on the house flies.

Reviewer 2 Report

General Comments

The paper by Saraswoti Neupane et al. titled “House flies (Musca domestica) pose a risk in carriage and transmission of bacterial pathogens associated with bovine respiratory disease (BRD)” reports that a significant number of house flies collected from a commercial feedlot from a pen with cattle suffering from apparent respiratory illness carried or contained one or more bacteria associated with BRD. The data are solid as the authors used both culture-dependent and independent methods to detect the presence of these bacteria. While a definitive statement about the vector competence of house flies for BRD pathogens cannot yet be made, the data suggest that house flies could potentially transmit BRD bacterial pathogens. I find this study very well executed, prepared and written, and I have only minor specific comments.

Specific Comments

(1)        Line 22: “apparent respiratory illness”. Is there some proof that cattle at the collection site indeed suffered from BRD?

(2)        I would liked to have already learned in the introduction how the flies acquire and carry these pathogens. This is mentioned on Line 175 in the discussion (“either on their surface or by ingestion”) but I kept wondering until then.

(3)        Line 41: manure? Do the authors mean “animal feces”? Manure is typically a “derived form”.

(4)        Line 83: Species identification was confirmed by…

(5)        Table 1: Not all columns are equally well aligned.

(6)        Table 2: The asterisk is difficult to spot. How about using a large size? As a table is a stand-alone unit, all abbreviations (CFU, BRD) should be spelled out in the caption or footnotes.

(7)        Line 138: “in eight total house flies”? Meaning is not clear. Please rephrase.

(8)        Line 142: Should the scientific names of these bacteria be italic?

(9)        Line 163: none of which was positive (please note: none=no one)

Author Response

Dear reviewer,

Thank you for your comments, we have followed all your suggestions and made the necessary corrections.  We have also added a comment into the introduction on the flies carry and disseminate the pathogens.

Reviewer 3 Report

The paper is a well written simply designed study of testing two samples of flies for several pathogens that contribute to a disease. The methods are clear and sound and the results tabulated well. My only major suggestion would be the addition, probably to the introduction but followed up on the results about the house flies vectoral ability and how it usually vectors pathogens and what you think might be the route here.

Small suggestions are detailed in the PDF attached.

Author Response

Dear reviewer,

thank you for your comments and corrections, we have followed them all with one exception. We could not find out what BBL abbreviation means in the GasPak system. It probably means just the label on the product.

Ludek